# Nano-Silver-Loaded Activated Carbon Material Derived from Waste Rice Noodles: Adsorption and Antibacterial Performance

**DOI:** 10.3390/nano14221857

**Published:** 2024-11-20

**Authors:** Guanzhi Ding, Guangzhi Qin, Wanying Ying, Pengyu Wang, Yang Yang, Chuanyang Tang, Qing Liu, Minghui Li, Ke Huang, Shuoping Chen

**Affiliations:** College of Materials Science and Engineering, Guilin University of Technology, Guilin 541004, China; 1020220195@glut.edu.cn (G.D.); 1020230210@glut.edu.cn (G.Q.); 2120210327@glut.edu.cn (W.Y.); 2120220368@glut.edu.cn (P.W.); 1020230205@glut.edu.cn (Y.Y.); 2120230405@glut.edu.cn (C.T.); 2120220336@glut.edu.cn (Q.L.); 2120230354@glut.edu.cn (M.L.); 3222042041416@glut.edu.cn (K.H.)

**Keywords:** waste rice noodles, nano-silver, activated carbon, adsorption, antibacterial activity

## Abstract

This study demonstrates, for the first time, the conversion of waste rice noodles (WRN) into a cost-effective, nano-silver-loaded activated carbon (Ag/AC) material capable of efficient adsorption and antibacterial activity. The fabrication process began with the conversion of WRN into hydrothermal carbon (HTC) via a hydrothermal method. Subsequently, the HTC was combined with silver nitrate (AgNO_3_) and sodium hydroxide (NaOH), followed by activation through high-temperature calcination, during which AgNO_3_ was reduced to nano-Ag and loaded onto the HTC-derived AC, resulting in a composite material with both excellent adsorption properties and antibacterial activity. The experimental results indicated that the incorporation of nano-Ag significantly enhanced the specific surface area of the Ag/AC composite and altered its pore size distribution characteristics. Under optimized preparation conditions, the obtained Ag/AC material exhibited a specific surface area of 2025.96 m^2^/g and an average pore size of 2.14 nm, demonstrating effective adsorption capabilities for the heavy metal Cr(VI). Under conditions of pH 2 and room temperature (293 K), the maximum equilibrium adsorption capacity for Cr(VI) reached 97.07 mg/g. The adsorption behavior of the resulting Ag/AC fitted the Freundlich adsorption isotherm and followed a pseudo-second-order kinetic model. Furthermore, the Ag/AC composite exhibited remarkable inhibitory effects against common pathogenic bacteria such as *E. coli* and *S. aureus*, achieving antibacterial rates of 100% and 81%, respectively, after a contact time of 4 h. These findings confirm the feasibility of utilizing the HTC method to process WRN and produce novel AC-based functional materials.

## 1. Introduction

As the most commonly used adsorbent, activated carbon (AC) plays a pivotal role in the removal of pollutants, water purification, and air management [1,2]. Modifications of AC, encompassing physical, acid, base, and loading modifications, can significantly enhance its adsorptive properties and endow it with other valuable functional characteristics. Among these, the strategy of loading specific substances onto the surface of AC can alter its specific surface area, pore size distribution, or other physicochemical properties [3]. This alteration enables the efficient adsorption of specific pollutants and the generation of high-value functional characteristics such as catalysis or antibacterial properties, which can significantly enhance the added value of the material [4]. The selection of the loading material is crucial for the loading modification of AC, as different loading materials can impart different properties or functions. Metallic particles, particularly nanoparticles, are one of the more common loading materials for AC [5,6,7,8]. For instance, Ramasundaram et al. synthesized silver nanoparticle-loaded cashew nut shell activated carbon (Ag/CNSAC), demonstrating that the synergistic effect of Ag as a photocatalyst and CNSAC as a catalytic carrier and adsorbent in Ag/CNSAC effectively degrades pollutants such as methylene blue, while also exhibiting excellent antibacterial efficiency against Escherichia coli and Staphylococcus aureus [6]. Chowdhury et al. initially synthesized AC from rice husk and then loaded Ni-Co-S (NCS) nanoparticles onto the AC surface through the reaction of metal precursors with thioacetamide at low temperatures, resulting in an AC-NCS composite material. Compared to unloaded AC and NCS, the AC-NCS exhibited significantly enhanced adsorption capacity for dyes and antibiotics [7]. Currently, the raw materials reported for loaded AC primarily derive from petroleum, coal, or biomass. Although these loaded ACs often exhibit outstanding functional characteristics, considering cost control, sustainable development, and waste-to-waste treatment strategies, loaded AC materials prepared from waste are clearly more environmentally valuable and competitive in the market.

As a significant category of municipal solid waste, catering waste poses a considerable ecological risk if it infiltrates soil and aquatic ecosystems, leading to severe environmental pollution and presenting substantial challenges for urban governance [9,10]. Traditional commercial catering waste treatment technologies include anaerobic digestion [11], aerobic composting [12], landfilling [13], incineration [14], and the production of animal feed [15]. While these strategies are capable of handling large volumes of food waste, they are associated with several evident drawbacks, such as the requirement for substantial land, capital, and equipment investments; low-value-added products and technological content; and the possible formation of secondary pollutants such as leachate or greenhouse gasses [16]. It is apparent that to achieve comprehensive and efficient resource utilization of catering waste, it is necessary to explore more effective supplementary approaches beyond conventional methods. These approaches should ensure that the products have the advantages of high technological content and high added value, and the production aspects should be characterized by simple operation, the absence of secondary pollution, low cost, and products with high added value.

Hydrothermal carbonization has gained significant attention in recent years as an innovative approach for the treatment of organic waste [17,18]. Through hydrothermal treatment, various natural organic materials, such as sugars [19], organic acids [20], fruit juices [21,22], and fruit peels [23,24], have been demonstrated to be convertible into new types of carbon-based functional materials like carbon quantum dots (CQDs) [25]. Compared to traditional methods of processing organic materials, hydrothermal carbonization offers several distinct advantages, including high conversion efficiency, minimal land requirement, reduced secondary pollution, and environmental friendliness [26]. In China, catering waste, which is rich in starch, could potentially be treated using hydrothermal carbonization strategies as a new avenue. Our preliminary research on waste rice noodles (WRN) has revealed that after hydrothermal treatment, the products can be divided into two parts based on their physical state: one part is liquid, containing a significant amount of CQDs, and the other is solid, referred to as hydrothermal carbon (HTC). These two fractions can be developed into different types of functional materials based on their distinct physicochemical properties. In our earlier work, we proposed, for the first time, a strategy for preparing novel photocatalytic composite materials from CQDs using WRN as a raw material. This strategy involves converting catering waste into CQDs through hydrothermal methods and then combining them with specific precursors to obtain CQD/TiO_2_ [27] or CQD/ZnO [28] photocatalytic composite powders for water pollution control, or even device-ready CQD/ZnO photocatalytic composite arrays based on catering waste [29]. Furthermore, we also achieved the synergistic transformation of two types of waste (WRN and waste iron oxide scale) into a magnetic CQD/FeO_x_ photocatalytic composite with extremely low cost and high recyclability, facilitating low-cost, large-scale photocatalytic water treatment [30]. On the other hand, we also conducted preliminary studies on the solid fraction produced from the hydrothermal treatment of WRN, known as HTC. HTC itself exhibited a small specific surface area and lacked effective adsorption capacity. However, it contained approximately 51% carbon and demonstrated significant reducing properties at high temperatures, indicating its potential for further derivatization to prepare novel carbon-based functional materials. Preliminary exploration revealed that under the presence of potassium hydroxide and following high-temperature activation, HTC could be converted into AC with a high surface area, showing some adsorption capacity for heavy metals such as hexavalent chromium [Cr(VI)] in water [27]. However, for complex real-world contaminated water, materials with only adsorption capabilities do not fully meet the requirements for efficient water treatment. For example, common AC typically shows limited effectiveness in inhibiting pathogenic microorganisms in water. Therefore, it is necessary to optimize the preparation strategy of the HTC pathway based on previous work, such as introducing suitable substances for loading, in order to develop high-value functional materials with enhanced adsorption capabilities and additional properties, such as antibacterial activity, to meet the practical requirements of wastewater purification.

Silver (Ag) nanoparticles have been demonstrated to be potent antimicrobial agents against a spectrum of Gram-positive and Gram-negative bacteria, and can be obtained through the reduction of silver salts [31,32,33]. Consequently, we propose a strategy for preparing a novel silver-loaded activated carbon from WRN via the HTC pathway, achieving, for the first time, the transformation of HTC into a high-value functional material with both excellent adsorption properties and antibacterial activity. This strategy involves the initial hydrothermal carbonization of WRN to obtain the HTC intermediate, followed by the mixture of HTC with an appropriate silver source (AgNO_3_) and an alkaline activator (NaOH). The mixture is then subjected to in situ activation and a complex reaction at high temperatures. This process not only forms AC with a high specific surface area, but also reduces AgNO_3_ on the HTC, resulting in the formation of silver nanoparticles loaded onto the AC’s surface. Ultimately, this leads to the creation of Ag/AC composite materials (see Figure 1). The morphology, composition, and structure of these Ag/AC materials are investigated, along with their adsorption capacity for Cr(VI) and their inhibitory effects on common pathogenic bacteria.

## 2. Experimental Section

### 2.1. Materials 

Similarly to our previous work [27,28,29,30], the catering waste, i.e., waste rice noodles (WRN), were collected from the canteen at Guilin University of Technology, located in Guilin, China. The silver nitrate (AgNO_3_, 99% purity), sodium hydroxide (NaOH, 96% purity), and potassium dichromate (K_2_Cr_2_O_7_, 98% purity) were purchased from Macklin Reagent (Shanghai, China) and used without further purification. The LB agar powder was purchased from Solarbio life sciences (Beijing, China). The *Escherichia coli* (*E. Coli*, ATCC 8739) and *Staphylococcus aureus* (*S. aureus*, ATCC 25923) were both obtained from the American Type Culture Collection (ACTT, Mansass, VI, USA).

### 2.2. Synthesis

Based on the findings from earlier experiments, we utilized a hydrothermal carbonization strategy to prepare Ag/AC composite materials using HTC as an intermediate [27,28,29,30]. Initially, 100 g of WRN was ground into a paste, which was then mixed with 200 g of deionized water, stirred evenly, and transferred into a 500 mL autoclave lined with Teflon (Kemi Instrument, Hefei, China). After sealing, the mixture was subjected to a hydrothermal reaction at 200 °C in an oven for 10 h, resulting in a dark brown solution with a black precipitate. Upon cooling, the solution was filtered, and the residue obtained was HTC. The HTC precipitate was placed in an oven and dried at 60 °C for 12 h, followed by grinding to give black HTC powder (yield: 19.2 g, 19.2% based on WRN). The filtrate, containing CQDs, could be processed through the CQD pathway to obtain various high-performance photocatalytic materials, thus achieving 100% recycling of WRN without causing secondary pollution [27,28,29,30].

Concurrently, 0.1 g of AgNO_3_ was dissolved in 5 mL of deionized water to form an AgNO_3_ solution, which was then mixed with 2 g of HTC powder and stirred evenly using a glass rod. The mixture was transferred to an oven and dried at 80 °C for 12 h. Subsequently, 6 g of NaOH was dissolved in 8 mL of deionized water to form a NaOH solution, which was combined with the silver-containing HTC from the previous step and mixed uniformly. The mixture was then dried in an oven at 80 °C for 48 h, yielding a black solid. The solid was subsequently placed in an SGM 6812 CK tubular furnace (Sigma, Luoyang, China) and heated at 900 °C for 1 h. After cooling to room temperature, the product was ground into a powder, washed with deionized water until neutral, filtered, and then transferred back to the oven to dry at 60 °C for 12 h, resulting in the final Ag/AC composite material. Following an identical preparation procedure, but omitting the introduction of AgNO_3_, an AC material without Ag loading was fabricated to serve as a control sample. The refinement of the experimental conditions for the Ag/AC composite is delineated in Appendix A.

### 2.3. General Characterization

The Ag/AC composite material was characterized using techniques described in our earlier studies [27,28,29,30]. For more comprehensive information, please see Appendix A.

### 2.4. Measurement of Adsorption Performance for Cr(VI) 

The adsorption performance of the Ag/AC composite for Cr(VI) was assessed using the procedures described in our earlier publications [27]. For further information, please consult Appendix A.

### 2.5. Measurement of Antibacterial Performance

The antibacterial properties of the Ag/AC material were assessed using a VersaMax microplate reader (Molecular Devices, Sunnyvale, NE, USA), with the AC material unloaded with silver serving as a control. For comprehensive details, please consult Appendix A.

## 3. Result and Discussion

### 3.1. Structural Characterization

AgNO_3_ is known for its oxidative properties, whereas HTC, rich in C and H elements, might exhibit reducibility at high temperatures. Consequently, at elevated temperatures, AgNO_3_ could be readily reduced to metallic Ag. The X-ray diffraction (XRD) pattern of the Ag/AC composite material demonstrated the presence of diffraction peaks at 2θ values of 38.12°, 44.28°, 64.43°, and 77.47°, which corresponded to the (111), (200), (220) and (311) crystal planes of cubic-phase Ag, respectively. These were in agreement with the standard diffraction card of Ag crystals (JCPDS card no. 04-0783), confirming that the phase loaded on the AC matrix surface was cubic-phase Ag. Due to the good crystallinity of the Ag phase and the strong diffraction signals, the AC diffraction peaks (broad peaks at 43° and 25°, representing the (100) and (002) crystal planes of graphite) were not pronounced (see Figure 2).

The SEM analysis of the Ag/AC composite revealed a typical porous morphology, with nanoscale Ag particles fairly uniformly dispersed on the surface of the AC (see Figure 3a). The EDS analysis also indicated that the Ag element was distributed quite evenly on the surface of the AC, which resulted in a Ag content of 2.05 wt% in the composite (see Figure 3b). The Ag/AC composite material was also subjected to ion thinning prior to TEM analysis. The results, as depicted in Figure 3c, revealed that nanoscale Ag was loaded onto the surface of the AC material and was closely integrated with the carbon substrate. The high-resolution TEM (HRTEM) image presented in Figure 3d further displays the interlacing of the lattices between Ag and the activated carbon support. Notably, the lattice fringes with interplanar spacings of 0.236 nm corresponded to the (111) crystal planes of Ag [34]. Lattice fringes with an interplanar spacing of about 0.309 nm were also observed, aligning with the (002) crystal plane of AC [35].

The FT-IR spectra of the Ag/AC composite material and the pristine AC are depicted in Figure 4a. It was observed that the AC surface, devoid of Ag loading, exhibited a substantial presence of hydroxyl groups, with the O–H stretching vibration peak positioned at 3420 cm^−1^, and the associated C–O stretching vibration was detected at 1049 cm^−1^. Upon Ag loading, the C–O stretching vibration peak of the Ag/AC composite material shifted to 1173 cm^−1^. This spectral shift suggests that during the high-temperature reduction of AgNO_3_ on HTC, a reaction likely occurred between the surface hydroxyl groups of AC and Ag ions, leading to the formation of C–O–Ag bonds, which is indicative of strong interfacial binding between the nanoscale Ag particles and the AC surface. Additionally, the point-of-zero-charge (PZC) measurement results revealed that the AC without Ag loading contained a higher concentration of hydroxyl groups, with its PZC at a lower pH value of 5.70. In contrast, after the formation of the Ag/AC composite, the surface hydroxyl groups were consumed, resulting in a significant elevation in the PZC for the Ag/AC composite material to 7.48 (see Figure 4b).

The successful integration of Ag with the AC surface was also evidenced by X-ray photoelectron spectroscopy (XPS). As depicted in Figure 4c, the XPS spectrum of the Ag/AC composite material revealed the presence of Ag, O, and C elements. The high-resolution spectrum of the Ag element exhibited signals at 368.64 eV and 374.63 eV, corresponding to the Ag 3d spectrum with Ag 3d_5/2_ and Ag 3d_3/2_ peaks (see Figure 4d and Table 1) [36]. In the high-resolution spectrum of O 1s, the Ag/AC composite material demonstrated the existence of C–O, O–H, and C–O–Ag bonding, with characteristic signals located at 530.67, 532.30, and 534.78 eV, respectively. In contrast, the O 1s spectrum of the pristine AC featured two characteristic peaks at 531.69 and 532.98 eV, corresponding to the oxygen in the C–O and O–H groups, respectively. In conjunction with the aforementioned IR spectroscopy results, it was inferred that the formation of the Ag/AC composite was actually a reaction between Ag ions and the hydroxyl groups on the AC surface, leading to the emergence of C–O–Ag bond signals (see Figure 4e). Furthermore, in the C 1s high-resolution spectrum, the characteristic peaks of the Ag/AC composite at 284.80, 286.43, and 288.89 eV were attributable to the C–C bonds, C–O bonds, and aromatic ring structures of C=C bonds within the composite material (see Figure 4f).

The nitrogen adsorption/desorption isotherms and pore size distribution experimental results, as shown in Figure 5 and Table 2, demonstrated that the incorporation of nanoscale Ag substantially enhanced the BET-specific surface area of the AC and significantly impacted its adsorption behavior. The AC without Ag loading exhibited a BET-specific surface area of 819.19 m^2^/g with an average pore size of 3.00 nm (see Table 2). Its adsorption isotherm could be interpreted as a mixture of Type I in the low-pressure range and Type IV in the medium- to high-pressure range, accompanied by an H4-type hysteresis loop, indicating its classification as a predominantly microporous and mesoporous adsorbent (see Figure 5b,d). Following the introduction of nanoscale Ag, the specific surface area was notably increased to 2025.96 m^2^/g, suggesting enhanced adsorptive capabilities. Simultaneously, the average pore size decreased to 2.14 nm, and the nitrogen adsorption profile closely approximated a Type I isotherm, indicating a shift toward a microporous adsorbent (see Figure 5a,c and Table 2).

### 3.2. Adsorption Performance for Cr(VI)

Given the significant specific surface area of the Ag/AC composite material, it was anticipated that this type of AC would effectively adsorb the heavy metal Cr(VI). At room temperature (293 K) and a pH of 2, the Ag/AC composite was subjected to a 24 h equilibrium adsorption study using Cr(VI) solutions with varying initial concentrations, and the results are shown in Figure 6a. The equilibrium adsorption data were fitted to both Langmuir and Freundlich isotherm models, with the linearized forms of the Langmuir and Freundlich equations presented in Figure 6b,c, respectively. The relevant parameters for the Langmuir and Freundlich models and their linear regression relationships are provided in Table 3. Based on the regression coefficients, the Langmuir model offered a better fit for describing the adsorption behavior of the material compared to the Freundlich model, suggesting that the adsorption active sites were uniformly distributed across the Ag/AC composite. According to the Langmuir model, the maximum equilibrium adsorption capacity of the Ag/AC composite for Cr(VI) reached 1.8667 mmol·g^−1^, equivalent to 97.07 mg/g. This adsorption capacity was higher than that of AC without Ag loading (65.43 mg/g; see Appendix A), as well as commercial coal-based AC (49.98 mg/g; see Appendix A). Furthermore, compared with various AC materials reported in the literature, the Ag/AC composite derived from WRN exhibited remarkable Cr(VI) adsorption performance, achieving an equilibrium adsorption capacity that exceeded those of Fe_2_O_3_/AC [37], Fe_3_O_4_/AC [38], and Ag-ZnO/AC [39], all synthesized from commercial AC (see Table 4). The Cr(VI) adsorption capacity of Ag/AC was also superior to that of the Al_2_O_3_/AC composite derived from vegetable crust leather waste [40], though it was slightly lower than that of ZnCl_2_-AC or K_2_CO_3_-AC composites prepared from tropical hardwood sawdust [41]. Therefore, the WRN-based Ag/AC composite represents a low-cost, high-performance solution with potential applications in heavy metal pollution control. It is also noteworthy that certain biomass-derived modified hydrochar materials, such as those prepared from bamboo [42] or poultry litter [43], demonstrate favorable Cr(VI) adsorption capabilities, with some hydrochars exhibiting photocatalytic Cr(VI) degradation [44]. However, for WRN-based systems, the directly hydrothermally synthesized HTC from WRN showed a maximum equilibrium Cr(VI) adsorption capacity of only 1.87 mg/g, rendering it impractical for real-world applications (see Appendix A). Thus, WRN-based HTC must be thermally activated at high temperatures and functionalized with additional loadings to serve as a high-value functional materials.

To further investigate the adsorption capacity and thermodynamics of the Ag/AC composite material for Cr(VI), adsorption isotherms were obtained at four different temperatures (293, 303, 313, and 323 K). As shown in Figure 7a and Table 3, the R^2^ values for the Langmuir model exceeded 0.999 at all four temperatures, significantly higher than those obtained from the Freundlich model. This further confirmed that the Langmuir model more effectively described the adsorption equilibrium of Cr(VI) on the Ag/AC composite material, indicating monolayer surface adsorption at specific homogeneous sites, as suggested by the Langmuir adsorption isotherms. A relationship between LnK_eq_ and 1/T was also established, yielding a correlation coefficient R^2^ of 0.9999, demonstrating a strong linear relationship (see Figure 7b). Table 5 presents the calculated thermodynamic values of ∆H, ∆S, and ∆G. It is apparent that the negative ∆G values gradually increased with decreasing temperature, indicating the spontaneity of the adsorption process. The positive ∆S suggested that the adsorption process was driven by entropy rather than enthalpy. Additionally, the positive ∆H value confirmed the endothermic nature of the adsorption process by the Ag/AC composite material, with higher temperatures favoring adsorption.

The WRN-based Ag/AC composite material also exhibited a favorable adsorption rate for Cr(VI). As shown in Figure 8a, equilibrium was achieved within 80 min for a 150 mg/L Cr(VI) solution. The experimental data were further analyzed using pseudo-first-order and pseudo-second-order kinetic models, with their linear forms shown in Figure 8b,c, respectively. The relevant fitting parameters and regression coefficients are detailed in Table 6. The results clearly show that the adsorption of Cr(VI) using the Ag/AC composite closely aligned with the pseudo-second-order kinetic model, suggesting that chemical adsorption was the main controlling factor.

On the other hand, SEM analysis revealed that after the adsorption of Cr(VI), the Ag/AC composite largely retained its porous morphology from before adsorption (see Figure 9a). EDS analysis indicated that the adsorbed Cr(VI) was relatively evenly distributed across the material’s surface (see Figure 9b). As shown in the PXRD analysis in Figure 9c, the adsorption of Cr(VI) did not lead to significant changes in the crystal structure of nano-silver, though the Ag diffraction peaks became slightly weaker. Furthermore, infrared spectroscopy showed the appearance of a Cr–O vibration peak at 486 cm^−1^ after Cr(VI) adsorption, indicating that some form of chemical interaction occurred between Cr(VI) and the Ag/AC composite during the adsorption process (see Figure 9d).

We also investigated the effect of different water types on the Cr(VI) adsorption capacity of the Ag/AC composite. As shown in Figure 10a, in a 100 mg/L Cr(VI) solution, the Ag/AC composite adsorbed 92.14% of Cr(VI) within 120 min in distilled water. However, under the same conditions, the Cr(VI) removal rate decreased to 68.13% and 61.20% in tap water and river water, respectively. This reduction could be attributed to the presence of anions such as sulfate, chloride, and phosphate ions in tap and river water, which likely competed with Cr_2_O_7_^2−^ ions, the main form of Cr(VI) in solution, for adsorption sites. Nonetheless, even in tap and river water, the Ag/AC composite demonstrated considerable Cr(VI) adsorption activity, suggesting its suitability for treating complex wastewater compositions. Additionally, the Ag/AC composite exhibited regenerability and reusability, as it could be regenerated by desorbing Cr(VI) through alkali leaching and then reused for Cr(VI) adsorption. As illustrated in Figure 10b, although the adsorption efficiency of the Ag/AC composite gradually decreased with repeated regeneration cycles, it still achieved a Cr(VI) removal rate of approximately 65% within 120 min after five adsorption cycles. This finding further supports the potential of Ag/AC composite as a cost-effective material for practical industrial water treatment applications.

### 3.3. Antibacterial Performance

The incorporation of nanoscale Ag significantly enhanced the adsorption performance of the Ag/AC composite material while also providing it with excellent antimicrobial properties. As illustrated in Figure 11, the AC without Ag loading exhibited only minimal bactericidal effects. After a 4 h contact period, the inhibition rate against *Escherichia coli* (*E. coli*, Gram-negative bacteria) was merely 22%, while the inhibition rate against *Staphylococcus aureus* (*S. aureus*, Gram-positive bacteria) was just 17%. Under the same conditions, the Ag/AC composite material achieved a 100% inhibition rate against *E. coli*, which is 4.5 times greater than that of the pristine AC, and an 81% inhibition rate against S. aureus, which is 4.7 times higher than that of the pristine AC. The antimicrobial efficacy conferred by the Ag loading enables the Ag/AC composite material to more effectively combat threats from bacteria and other microorganisms in wastewater, making it more suitable for efficient, industrial-scale water purification processes. Additionally, it was observed that, under identical conditions, the Ag/AC composite exhibited stronger antibacterial activity against *E. coli* than against *S. aureus*. This effect could be attributed to the structural differences between these bacteria. As a Gram-negative bacterium, *E. coli* possessed only a thin peptidoglycan layer, covered by an outer lipid bilayer membrane. The Ag nanoparticles on the Ag/AC composite, along with the released Ag^+^ ions, were more readily able to penetrate this relatively weak barrier and enter the cell, leading to protein inactivation, DNA damage, and metabolic disruption. In contrast, *S. aureus*, a Gram-positive bacterium, had a thick peptidoglycan layer in its cell wall and lacked an outer membrane. This thick peptidoglycan layer partially inhibited the penetration of silver, making it more difficult for Ag^+^ ions to enter the cell rapidly or in significant amounts. This structural protection rendered *S. aureus* relatively more resistant to the Ag/AC composite [47].

## 4. Conclusions

This study successfully utilized WRN as a raw material to synthesize a low-cost, multifunctional Ag/AC composite material through the HTC pathway, incorporating in situ reduction. The experimental results demonstrated that the incorporation of nanoscale Ag significantly increased the specific surface area of the material, thereby enhancing the adsorption performance of the Ag/AC composite for heavy metals such as Cr(VI), while also providing commendable antibacterial capabilities, which confer a distinct advantage for efficient water purification.

Additionally, our findings preliminarily validated the feasibility of developing multifunctional, high-value water purification materials via the HTC approach. Compared to traditional methods of treating food waste, the HTC pathway and its resultant Ag/AC products may offer significantly higher profit margins and added value than mainstream food waste products. This method presents broader application scenarios and stronger market competitiveness, while also being more convenient and environmentally friendly in terms of production processes, requiring less investment in equipment and facilities. Furthermore, this approach can complement previous work reported on CQDs, which enable the production of metal oxide/CQD composite materials and devices, achieving 100% utilization of WRN and preventing secondary pollution, thus providing a novel pathway for recycling food waste and preparing low-cost, high-performance functional materials from food waste.

## Figures and Tables

**Figure 1 nanomaterials-14-01857-f001:**
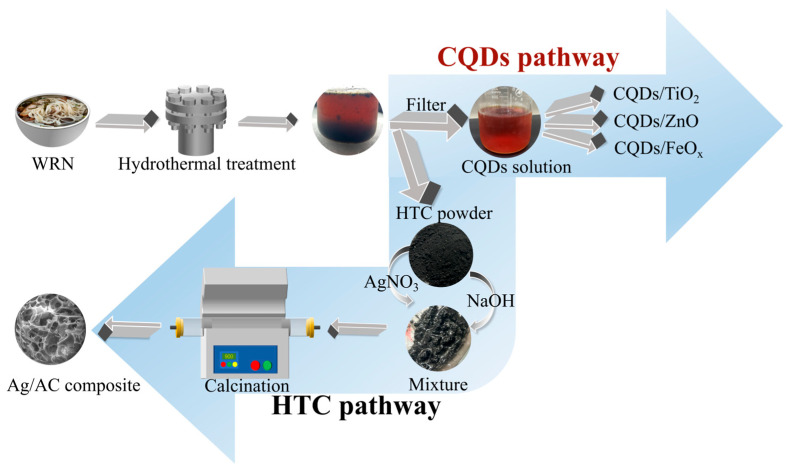
Process flow diagram for simultaneous preparation of Ag/AC composite and photocatalytic composite (such as CQD/TiO_2_ [27], CQD/ZnO [28,29] and CQD/FeO_x_ [30]) using WRN as raw material.

**Figure 2 nanomaterials-14-01857-f002:**
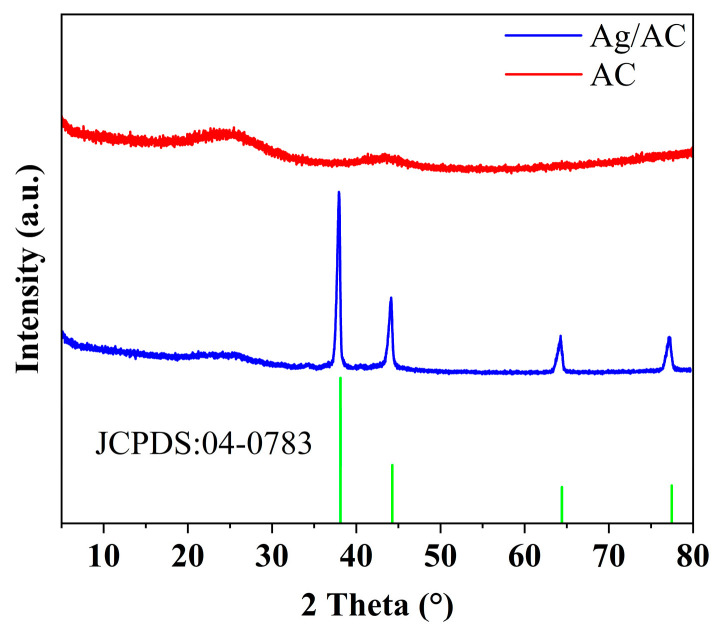
PXRD pattern of AC and Ag/AC composite.

**Figure 3 nanomaterials-14-01857-f003:**
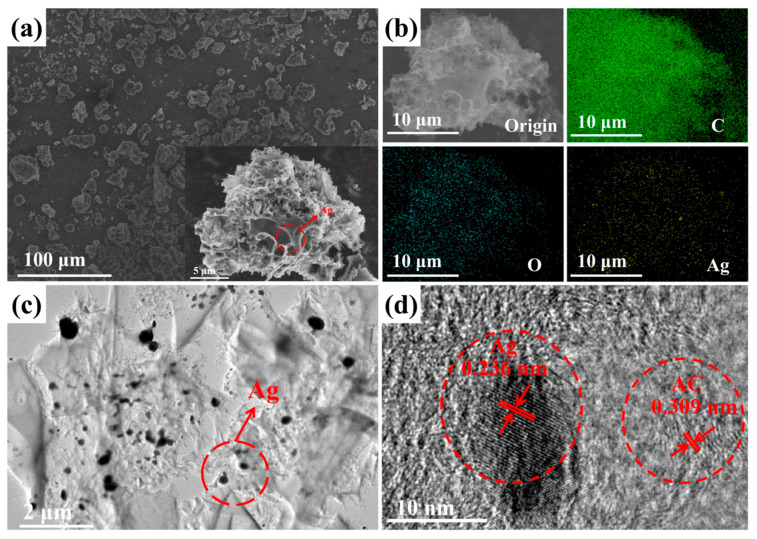
(**a**,**b**) The SEM image (**a**) and elemental mapping (**b**) of the Ag/AC composite based on WRN; (**c**,**d**) the TEM image (**c**) and HRTEM image (**d**) of the Ag/AC composite based on WRN.

**Figure 4 nanomaterials-14-01857-f004:**
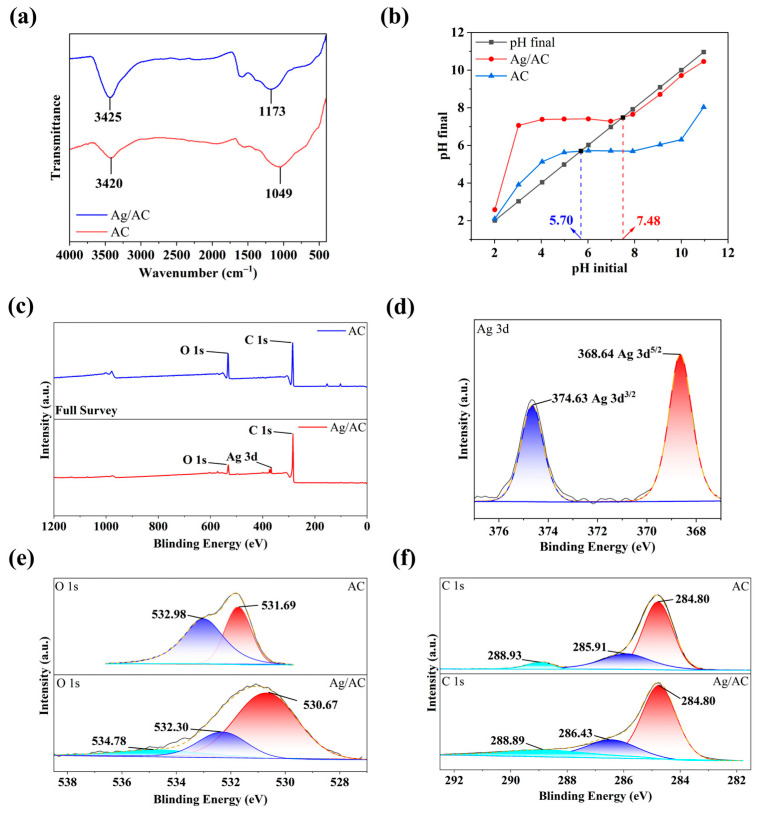
(**a**) IR spectra of AC and Ag/AC composite. (**b**) PZC value of AC and Ag/AC composite; (**c**–**f**) Full XPS (**c**), Ag 3d ((**d**), Ag/AC composite only), O 1s (**e**), and C 1s (**f**) high-resolution spectrum of AC and Ag/AC composite.

**Figure 5 nanomaterials-14-01857-f005:**
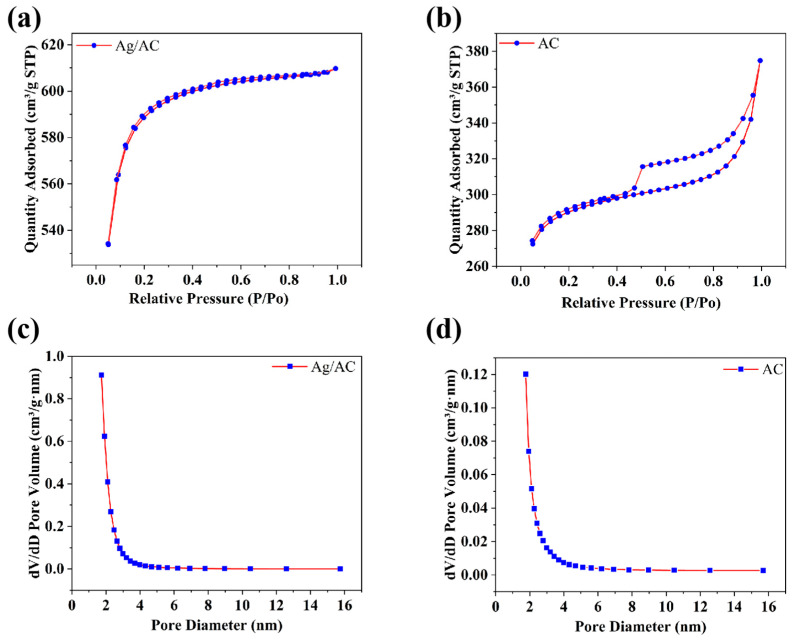
(**a**,**b**) Nitrogen adsorption/desorption isotherms of Ag/AC composite (**a**) and AC without Ag loading (**b**). (**c**,**d**) Pore size distribution curves of Ag/AC composite (**c**) and AC without Ag loading (**d**).

**Figure 6 nanomaterials-14-01857-f006:**
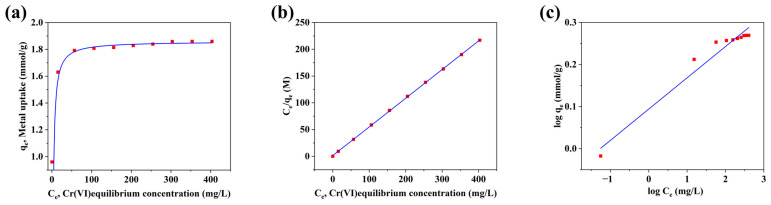
Adsorption isotherm of Ag/AC composite for Cr(VI): (**a**) experimental data, (**b**) Langmuir model, and (**c**) Freundlich model.

**Figure 7 nanomaterials-14-01857-f007:**
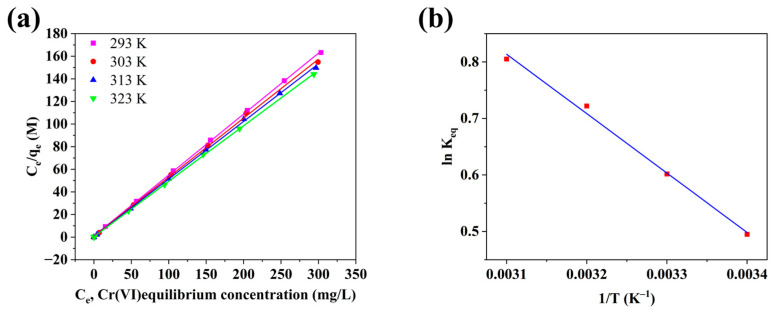
(**a**) Langmuir adsorption isotherms of Cr (VI) on Ag/AC composite materials at different temperatures. (**b**) Relationship between ln k_eq_ and 1/T.

**Figure 8 nanomaterials-14-01857-f008:**
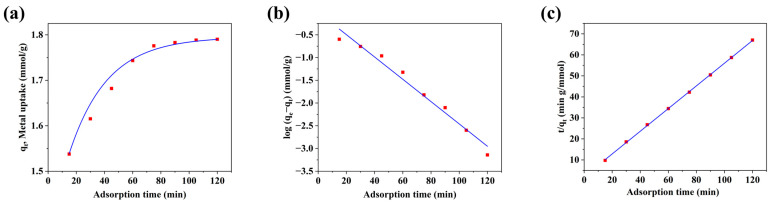
Dynamic adsorption equilibrium curve (**a**), fitted pseudo-first-order kinetic model (**b**) and pseudo-second-order kinetic model (**c**) of Ag/AC composite.

**Figure 9 nanomaterials-14-01857-f009:**
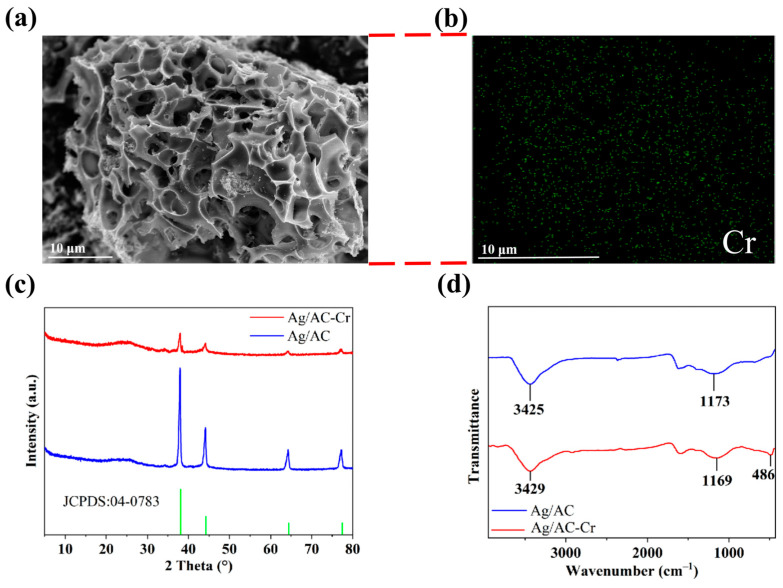
(**a**,**b**): SEM image (**a**) and Cr(VI) elemental mapping (**b**) of Ag/AC composite after Cr(VI) adsorption; (**c**) PXRD patterns of Ag/AC composite before and after adsorption; (**d**) IR spectra of Ag/AC composite pre- and post-adsorption. Samples were treated in 150 mg·L^−1^ Cr(VI) solution with 1 g·L^−1^ adsorbent for 2 h.

**Figure 10 nanomaterials-14-01857-f010:**
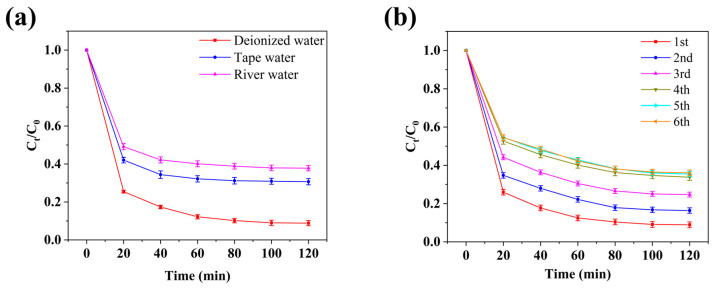
(**a**) Adsorption efficiency of Ag/AC composite for Cr(VI) in different water types. (**b**) Adsorption performance of Ag/AC composite for Cr(VI) after multiple adsorption–regeneration cycles.

**Figure 11 nanomaterials-14-01857-f011:**
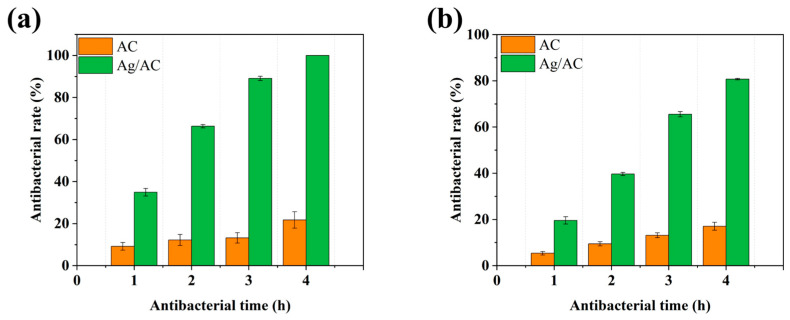
(**a**,**b**) Inhibition rates of AC and Ag/AC composite materials against *E. coli* (**a**) and *S. aureus* (**b**) for different times.

**Table 1 nanomaterials-14-01857-t001:** XPS peak distribution of AC and Ag/AC composite based on WRN.

	Element	Peak (eV)	Surface Group	Assignment
AC	C 1s	284.80	C	Graphitic carbon
		285.91	C–O	Alcoholic or etheric structure in AC
		288.93	C=C	π-electrons in aromatic ring
	O 1s	531.69	C–O	Oxygen atom bonded to aromatic rings
		532.98	O–H	Hydroxyl group
Ag/AC	C 1s	284.80	C	Graphitic carbon
		286.43	C–O	Alcoholic or etheric structure in AC
		288.89	C=C	π-electrons in aromatic ring
	O 1s	530.67	C–O	Oxygen atom bonded to aromatic rings
		532.30	O–H	Hydroxyl group
		534.78	C–O–Ag	Oxygen bonded to silver
	Ag 3d	368.64	Ag	Ag 3d_5/2_
		374.63	Ag	Ag 3d_3/2_

**Table 2 nanomaterials-14-01857-t002:** Specific surface area and pore characteristic of Ag/AC composite and AC without Ag loading.

Sample	S_BET_ (m^2^/g)	S_mic_ (m^2^/g)	V_tot_ (cm^3^/g)	V_mic_ (cm^3^/g)	D_ave_ (nm)
AC	819.19	740.98	0.58	0.38	3.00
Ag/AC	2025.96	966.73	0.56	0.48	2.14

**Table 3 nanomaterials-14-01857-t003:** Langmuir and Freundlich isotherm adsorption parameters for Cr(VI) at different temperatures.

T (K)	Langmuir Models			Freundlich Models		
	q_m_ (mmol·g^−1^)	b (L·mg^−1^)	R^2^	k_f_ (mmol·g^−1^)	1/n	R^2^
293	1.8667	0.399	0.9998	1.2429	0.073	0.9461
303	1.9168	0.591	0.9993	1.7317	0.166	0.7953
313	1.9657	0.850	0.9996	1.8181	0.013	0.7772
323	2.0370	1.534	0.9998	1.9037	0.012	0.9261

**Table 4 nanomaterials-14-01857-t004:** Adsorption capacity of Cr(VI) by AC and related adsorbents from various raw materials.

Adsorbent	Carbon Source	Metal Source	Adsorbent Dosage (g·L^−1^)	pH	Adsorption Capacity (mg·g^−1^)	Reference
AC	Banana peels, corn cobs	-	0.4	2	19.16	[23]
AC	Mango kernel	-	2.5	2	7.8	[21]
AC	Aloe vera waste leaves	-	2	2	58.83	[45]
AC	Date press cake	-	1	5	198	[46]
AC	Hard shell of wood apple fruit	-	1.25	2	151.51	[24]
Zn-modified hydrochar	Bamboo	ZnCl_2_	3.3	5	14.0	[42]
Al-modified hydrochar	Bamboo	AlCl_3_	3.3	5	12.3	[42]
H_2_SO_4_-modified hydrochar	Poultry litter	-	2	2	26.21	[43]
ZnCl_2_-AC	Tropical hardwood sawdusts of *Tectona grandis* tree	ZnCl_2_	0.4	3	127	[41]
K_2_CO_3_-AC	Tropical hardwood sawdusts of *Tectona grandis* tree	K_2_CO_3_	0.4	3	103	[41]
Fe_2_O_3_/AC	Commercial AC	FeSO_4_·5H_2_O FeCl_3_·6H_2_O	1	2	83.33	[37]
Fe_3_O_4_/AC	Commercial AC	FeSO_4_·7H_2_O FeCl_3_·6H_2_O	2	2	45.3	[38]
Al_2_O_3_/AC	Vegetable crust leather waste	Al_2_O_3_	6	6	19.3	[40]
Ag-ZnO/AC	Commercial AC	Zn(NO_3_)_2_ AgNO_3_	16	2.5	4.17	[39]
Ag/AC	WRN	AgNO_3_	1	2	97.07	This work
AC	WRN	-	1	2	65.43	This work
Commercial AC	Coal	-	1	2	49.98	This work
HTC	WRN	-	1	2	1.87	This work

**Table 5 nanomaterials-14-01857-t005:** The relationship between ∆H, ∆S, and ∆G in thermodynamics.

T (K)	ΔH (kJ·mol^−1^)	ΔS (kJ·mol^−1^·K^−1^)	ΔG (kJ·mol^−1^)
293	9.582	0.03652	−1.186
303			−1.524
313			−1.863
323			−2.201

**Table 6 nanomaterials-14-01857-t006:** The kinetic adsorption parameters for the adsorption of Cr(VI) obtained using pseudo-first-order and pseudo-second-order models.

Pseudo-First-Order			Pseudo-Second-Order		
q_e1_ (mmol·g^−1^)	K_1_ (min^−1^)	R_1_^2^	q_e2_ (mmol·g^−1^)	K_2_ (g·mmol·min^−1^)	R_2_^2^
0.9883	0.0566	0.9751	1.8513	0.1436	0.9997

## Data Availability

All data are included in the article and Appendix A.

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
