# Peer review of "Nano-Silver-Loaded Activated Carbon Material Derived from Waste Rice Noodles: Adsorption and Antibacterial Performance"

_nanomaterials, 2024, doi:10.3390/nano14221857_

Round 1
Reviewer 1 Report
Comments and Suggestions for Authors
This manuscript reported the preparation of a composite obtained from an hydrochar loaded with silver nanoparticles. The composite is applied for Cr(VI) removal and antibacterial activity. The antibacterial activity is tested on Gram+ and Gram- bacteria and the antibacterial activity was higher with Gram- bacteria. The Cr(VI) removal is evaluated by adsorption in batch mode and the performance are well discussed and compared with literature. The manuscript is well written and the discussion is correlated to experimental data. The scientific goal is interesting because the authors demonstrate that rice waste could be valorized by hydrothermal treatment in solid and liquid phase. The liquid phase could be converted in carbon quantum dot (previous work) while solid phase could be valorize in activated carbon. (this work).
So, I suggest accepting the manuscript after major revisions.
1) In the experimental part, the authors should give the weight of HTC obtained from 100 g of WRN to have any idea of the conversion rate.
2) In synthesis, the authors indicate that optimization of experimental conditions are presented in Table S1 and table S1. One of table S1 has to be erase.
3) In part 3.1, figure citations have to be checked because Figure 4a and 4b didn’t correspond to the sentences.
4) In part 3.2, the authors compared the adsorption performance with literature and demonstrated that the material is relevant with commercial activated carbon obtained
5) In table 3, the authors should include the adsorption capacity in mg/g for HTC and AC without loading to discuss about the performance of the unloaded material compared to other HTC or AC samples. References presented in table 3 are interesting and illustrated that the performance of the AG/AC composite is interesting. However, references from Luo et al. (https://doi.org/10.1016/j.seppur.2023.123926); (https://doi.org/10.1016/j.chemosphere.2020.127610), Vo et al. (https://doi.org/10.3390/w11061164.), and Ghanim et al.(https://doi.org/10.1002/jctb.6904) have to be include to complete the discussion.
6) In the discussion, the authors evaluate the performance in batch mode with a solution of distilled water doped with chromium. To be relevant, the composite has to be tested with complex solution such as tap water, river water, or groundwater.
7) In the antibacterial part, the authors should add few sentences to explain why Gram- bacteria are more sensitive to the presence of composite.

Reviewer 2 Report
Comments and Suggestions for Authors
This study proposes a strategy to convert waste rice noodles (WRN) into a cost-effective, nano silver-loaded activated carbon (Ag/AC) with excellent adsorption and antibacterial properties. The process involves converting WRN into hydrothermal carbon (HTC) via hydrothermal treatment, followed by combining HTC with silver nitrate (AgNO₃) and sodium hydroxide (NaOH), and activating it through high-temperature calcination.
The manuscript contains interesting data. Generally, the results are well discussed providing consistent explanations. I recommend its publication in Nanomaterials after minor corrections:
- In the Introduction section the authors should highlight more the novelty of the study!
-The discussion regarding the porosity of the samples should be revised! In my opinion the isotherm of the AC without Ag loading (Fig. 5b) seems to be a hybrid of type I (in the range of low relative pressures) and type IV (in the range of moderate and high pressures; formation of hysteresis loops). Additionally, in the pore size distribution (PSD) graph (Fig. 5d) the intense peak centered at ~ 4 nm is an artifact due to the tensile strength effect. The PSD graph derived from the desorption branch is affected by pore network effects leading to a forced closure of the hysteresis loop. In such a case, the adsorption branch is highly preferred for pore size calculations because this branch is hardly affected by tensile strength effect.
-A table containing the calculated textural parameters (specific surface area, total pore volume, micropore volume, micropore area) should be given and discussed.
- Regarding the adsorption experiments, it would be interesting to complete the manuscript with a study on the regeneration and reusability of these materials.
